# Peripheral Blood Mononuclear Cells from Patients with Type 1 Diabetes and Diabetic Retinopathy Produce Higher Levels of IL-17A, IL-10 and IL-6 and Lower Levels of IFN-γ—A Pilot Study

**DOI:** 10.3390/cells12030467

**Published:** 2023-01-31

**Authors:** Gideon Obasanmi, Noemi Lois, David Armstrong, Jose M. Romero Hombrebueno, Aisling Lynch, Mei Chen, Heping Xu

**Affiliations:** 1Wellcome-Wolfson Institute for Experimental Medicine, School of Medicine, Dentistry and Biomedical Sciences, Queen’s University Belfast, Belfast BT9 7BL, UK; 2Department of Ophthalmology and Visual Sciences, Faculty of Medicine, University of British Columbia, Vancouver, BC V5Z 3N9, Canada; 3Institute of Inflammation and Ageing, College of Medical and Dental Sciences, University of Birmingham, Birmingham B15 2TT, UK

**Keywords:** type 1 diabetes, diabetic retinopathy, peripheral blood mononuclear cells, plasma, flow cytometry, intracellular cytokines

## Abstract

Inflammation is key to the pathogenesis of diabetic retinopathy (DR). This prospective study investigated alterations in inflammatory cytokines in peripheral blood mononuclear cells (PBMCs) in 41 people with type 1 diabetes (T1D), sub-grouped into mild non-proliferative DR (mNPDR; *n* = 13) and active and inactive (each *n* = 14) PDR. Age/gender-matched healthy controls (*n* = 13) were included. PBMCs were isolated from blood samples. Intracellular cytokine expression by PBMCs after 16-h stimulation (either *E. coli* lipopolysaccharide (LPS), phorbol 12-myristate 13-acetate plus ionomycin, D-glucose or D-mannitol) were assessed by flow cytometry. Cytokine production in plasma, non-stimulated and LPS-stimulated PBMC supernatant was also assessed. Increased BMC IL-10 secretion and reduced expression of IL-6 and IFN-γ in CD3^+^ cells were observed in mNPDR. Reduced IL-6 and IL-10 secretion, and higher levels of intracellular IL-6 expression, especially in CD11b^+^ PBMCs, was detected in aPDR; levels were positively correlated with DR duration. Patients with T1D demonstrated increased intracellular expression of IL-17A in myeloid cells and reduced IFN-γ expression in CD3^+^ cells. Plasma levels of IL-1R1 were increased in mNPDR compared with controls. Results suggest that elevated PBMC-released IL-10, IL-6, in particular myeloid-produced IL-17A, may be involved in early stages of DR. IL-6-producing myeloid cells may play a role in PDR development.

## 1. Introduction

Diabetic retinopathy (DR) is one of the most prevalent diabetic complications and a leading contributor to vision loss and blindness in the adult working population [1,2]. The detailed pathogenesis of DR is unclear, but it is accepted that inflammation is a key contributory factor to its multifactorial pathology [3,4,5]. This is supported by overwhelming evidence wherein infiltrating leukocytes, including peripheral blood mononuclear cells (PBMCs) and the cytokines which they release, are involved in DR pathological changes including increased leukostasis, capillary degeneration, neurodegeneration, vascular permeability and atypical immune responses [3,4,5,6,7,8,9].

Cytokines are a class of proteins produced by different types of cells—mostly leukocytes either constitutively or after activation; they function as molecular mediators of the innate and adaptive immunities, typically serving as intermediaries within and between these subsystems. The families of cytokines include chemokines which induce chemotaxis; interleukins (including most lymphokines) which perform various functions including maturation and proliferation of immune cells; interferons which function mainly to resolve pathogenic presence; as well as the tumor necrosis factor (TNF-α) family which regulates immune cell functions [10,11]. Various cytokines have been implicated in DR pathophysiology, as high concentrations of leukocyte-derived cytokines and growth factors including IL-1β, TNF-α, IL-1, IL-2, IL-6 and IL-8 have been reported in the vitreous humor, serum and plasma of DR patients [12,13,14,15,16]. In diabetes, a hyperglycemic microenvironment inundated with dysregulated cytokine secretion and expression is a recipe for low-grade cellular activation and inflammation (parainflammation) that may persist to prompt the onset of diabetic complications, including DR and its progression [17,18].

We previously reported that DR in type-1-diabetes (T1D) is associated with increased innate and reduced adaptive cellular immunity [19]. Despite such altered immunophenotype, we lack detailed information on how PBMC-cytokine production is altered in people with T1D at different stages of DR (early and advanced). In this follow-up study, we investigated whether cytokine production and secretion profiles of innate (CD11b^+^/myeloid cells) and adaptive (CD3^+^) peripheral blood immune cells are dysregulated in people with T1D with DR. Furthermore, we investigated whether PBMC cytokine secretion and expression is altered due to treatment in patients with advanced DR (proliferative diabetic retinopathy (PDR)). Uncovering this information is critical, not only to improve our understanding of DR pathophysiology, but also to design more effective therapies capable of normalizing the altered immune response underpinning DR. Furthermore, an association between diabetes’ duration and DR initiation and progression has been established [20,21,22,23,24], but the potential relationship between diabetes’ standing and specific PBMC-expressed, PBMC-secreted and/or plasma-expressed cytokine biomarkers have not been investigated in people with T1D with DR.

Thus, herein, we determine cytokine secretion and expression profiles of different subsets of PBMCs under normal and stimulatory culture conditions as well as the plasma cytokine profiles in patients with T1D and early and advanced DR and sought to understand whether these relate to diabetes’ duration. The stimulatory conditions included (1) high glucose (HG) to mimic hyperglycemia, (2) mannitol to mimic hypertonicity and serve as osmotic control for HG, (3) phorbol myristate acetate (PMA) and Ionomycin to mimic inflammatory conditions and (4) lipopolysaccharide (LPS) to mimic the endotoxin-induced inflammatory environment.

## 2. Materials and Methods

### 2.1. Study Participants

The study was approved by the Office for Research Ethics Committees Northern Ireland (ORECNI, Ref: 14/NI/0084) and was conducted in accordance with the Declaration of Helsinki. All participants were informed about the study and provided written informed consent prior to their enrolment. Forty-one adults with T1D (Age ≥ 18) were recruited into three groups: 13 T1D with mild non-proliferative DR (mNPDR), 14 T1D with active proliferative DR (aPDR) naïve to treatment and 14 T1D with PDR that had received laser PRP and were clinically stable and quiescent afterwards (inactive PDR; iPDR). Thirteen age- and gender-matched healthy controls were also recruited. The diagnosis of mNPDR, aPDR, iPDR in T1D and of no DR in healthy controls was made based on clinical history and clinical examination, including fundus photography. Medical and family history, current medications, body mass index (BMI) and systolic and diastolic blood pressure were obtained. Exclusion criteria included history of cardiac disease or malignancy within the past 5 years; history of inflammatory diseases within the past 2 months; and other retinal disorders besides DR, active autoimmune disease and history or current use of immunosuppressive medications or steroids. Pregnant females, people with kidney failure and incapacity to undertake eye imaging due to any reason were also excluded from the study.

### 2.2. Masking

Researchers who carried out laboratory experiments and analysis were masked to the origin of the clinical samples to be analyzed (i.e., whether they were coming from healthy volunteers or from patients with T1D and DR).

### 2.3. PBMC Isolation, Culture and Supernatant Collection

Venous blood (20 mL) was collected into tubes containing ethylenediaminetetraacetic acid (ETDA; BD Biosciences, Oxford, UK). Within 3 h of blood collection, PBMC were isolated by Ficoll-Paque (Histopaque; Sigma-Aldrich, Cambridge, UK) density gradient centrifugation. The PBMCs were cultured in RPMI 1640 medium containing 10% FCS and 1% penicillin-streptomycin under normoxia (21% oxygen); and incubated with either *Escherichia coli* lipopolysaccharide (LPS; 2.5 µg/mL), or PMA (100 ng/mL) + ionomycin (1 µg/mL), or 25 mM D-glucose, or 25 mM D-mannitol (all from Sigma–Aldrich(Cambridge, UK)) for 16 h. The supernatants were collected and stored at −80 °C until analysis.

### 2.4. Plasma Collection

After blood collection, the plasma was isolated by centrifugation of samples at 1200 rpm for 10 min at room temperature (RT). The plasma fraction was centrifuged again at 3000 rpm for 15 min at RT to pellet any residual cells and platelets. The samples were aliquoted and stored at −80 °C until analysis.

### 2.5. Protein Measurements in Plasma and PBMC Culture Supernatant

Soluble CD121a (sCD121a), IL-6, IL-8 and MCP-1 expression in plasma, and expression of cytokines: IL-1α, IL-1β, IL-6, IL-8, IL-10, IL-17A, IFN-γ, MCP-1 and TNF-α in culture supernatants of PBMCs with or without LPS stimulation were measured using Cytometric Bead Array Human Soluble Protein Detection Kit (CBA; BD Biosciences) per the manufacturer’s instructions. Briefly, capture beads coated with anti-cytokine antibodies were mixed and incubated with plasma or PBMC supernatant or standards in round bottom 96-well plates (Nunc; Thermo Scientific, Leicestershire, UK) for 1 h at RT in the dark. This was followed by an additional 2 h of incubation with phycoerythrin (PE)-conjugated anti-cytokine detection reagent at RT in the dark. The samples were washed, and the bead pellets were re-suspended in washing buffer. The resuspended samples were then run on a flow cytometer (FACS Canto II; BD Biosciences) that was equipped with BD Diva software. Collected data were subsequently analyzed using BD FCAP Array software version 3 (BD Biosciences). The total protein concentration per sample was measured using a Pierce BCA protein assay kit (Thermo Fisher Scientific, Waltham, MA, USA). The concentration of each cytokine was normalized to the total protein concentration in supernatant (pg/mg total protein).

### 2.6. PBMC Flow Cytometry: Surface Markers and Intracellular Cytokines

PBMCs cultured for 16h with different stimulants as described above were incubated alongside 1× monensin (Biolegend UK Ltd. London, UK) for 4 h. Monensin is a protein transport inhibitor that prevents cytokine secretion, thereby trapping them intracellularly. PBMCs were washed twice with FACS buffer (300× *g*, 4 °C for 7 min) and resuspended at 10 × 10^6^ cells/mL; 20 µL (2 × 10^5^ cells) of single cell suspension were allotted per FACS tube and incubated with 5 μL Human TruStain FcX (Fc Receptor blocking solution; BioLegend UK Ltd.) for 5 min at RT.

The cells were then incubated with cell surface fluorochrome-conjugated antibodies (See Table 1) in a total volume of 100 μL FACS buffer for 30 min in the dark at 4 °C. Post-fluorochrome staining, cells were washed twice with FACS buffer and then fixed and permeabilized using the Foxp3 Transcription Factor Staining Buffer Set (eBioscience, San Diego, CA, USA) according to the manufacturer’s instructions. Samples were then incubated with 5 μL Human TruStain FcX (Fc Receptor blocking solution; BioLegend UK Ltd.) followed by incubation with either intracellular cytokine fluorochrome-conjugated antibodies or appropriate isotype controls (see Table 1) in a total volume of 100 μL permeabilization buffer (eBioscience) for 40 min in the dark at 4 °C. Cells were washed and acquired on the FACSCanto II flow cytometer (BD Biosciences). Data analysis was performed using FlowJo software version 10.07 for Windows (Tree Star, Ashland, OR, USA).

### 2.7. Flow Cytometry Gating Strategies

Flow cytometry confirmed the depletion of granulocytes in isolated PBMCs (Figure 1a, right panel). Lymphocytes and myeloid-derived cells were identified by their surface expression of CD3 or CD11b (Figure 1b) as well as using an FSC vs. SSC dot-plot to gate on PBMC myeloid cells (Figure 1a, right panel). To analyze the expression of intracellular cytokine expression, gates of intracellular cytokines (e.g., IL-6, IL-17A, IL-10 and IFN-γ) were set on total live cells (Figure 1c–f), as well as on CD11b^+^ (Figure 1g), PBMC myeloid cells (Figure 1h–j) and CD3^+^ (Figure 1k–m) based on the appropriate isotype controls.

### 2.8. Statistical Analyses

Statistical calculations were performed with GraphPad Prism 6 (GraphPad, San Diego, CA, USA). Normality of continuous variables was determined using D’Agostino–Pearson omnibus normality test. Comparisons between two normally distributed groups were carried out using independent samples Student’s *t*-test while the Mann–Whitney test was used when normality was not confirmed. Linear regression analysis was used to investigate potential associations between T1D duration and PBMC cytokine expression. Data were presented as mean ± standard error of the mean (SEM) in figures and mean ± standard deviation (SD) in tables. *p* values < 0.05 were considered statistically significant.

**Figure 1 cells-12-00467-f001:**
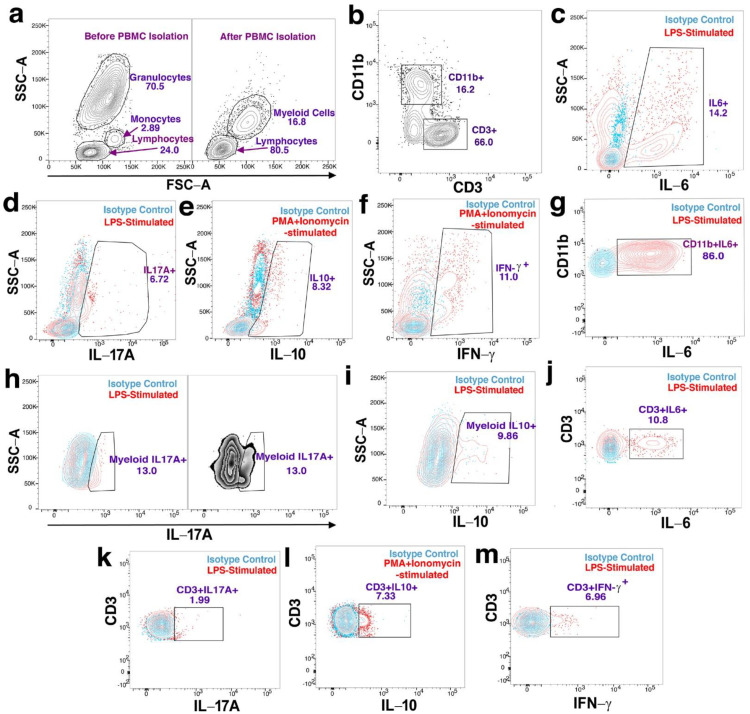
Gating strategies used in flow cytometry analysis to identify PBMC subsets and quantitate cytokine expression. (**a**,**b**) Representative data showing PBMC isolation and subsets: (**a**) Forward scatter (FSC−A) vs. side-scatter (SSC−A) plot of whole blood before (Left panel) and after PBMC isolation (Right panel), note absence of polymorphonuclear granulocytes. (**b**) CD3 vs. CD11b showing CD3^+^ and CD11b^+^ PBMCs. (**c**–**f**) Representative data used to quantitate overall cytokine expression via cytokine vs. SSC−A plots to indicate the PBMC expression of: (**c**) IL-6; (**d**) IL-17A; (**e**) IL-10 and (**f**) IFN-γ. (**g**–**i**) Representative data used to quantitate cytokine expression in the CD11b^+^ and myeloid cell subsets of PBMCs via cytokine vs. CD11b^+^ or SSC−A plots: (**g**) IL-6; (**h**) IL-17A (Left panel shows myeloid cell IL-17A expression in contour display while right panel shows IL-17A expression without isotype control overlay in zebra display) and (**i**) IL-10. (**j**–**m**) Representative data used to quantitate cytokine expression in the CD3^+^ subsets of PBMCs via cytokine vs. CD3 plots: (**j**) IL-6; (**k**) IL-17A; (**l**) IL-10 and (**m**) IFN-γ. Populations in blue color have been stained with isotype controls while populations in red color have been stained with dye-conjugated antibodies. The gate frequencies displayed on plots are the percentages of events within each gate.

## 3. Results

### 3.1. Demographics and General Information of Study Participants

We did not find statistically significant differences in demographics (age, gender) or systolic or diastolic blood pressure (SBP, DBP, respectively), BMI, or family history of T1D, smoking history, time of blood collection between patients with T1D and DR and healthy controls. Detailed information on characteristics of the participants has been published elsewhere [19]. Briefly, females represented 29.6% of the total 54 recruited subjects, including 12 female with diabetic retinopathy (DR) (22.2% of DR patients) with a breakdown of 4 with mNPDR (30.8% of mNPDR patients), 5 with inactive DR (35.7% of iPDR patients), and 3 with active PDR (21.4% of aPDR patients), as well as 4 healthy female controls (30.8% of healthy controls). The mean age ± standard deviation (years) for all subjects was 47 ± 11.7, including 47.8 ± 12.27 for DR patients, 47.8 ± 9.2 for mNPDR patients, 41.9 ± 12.5 for iPDR and 53.6 ± 12.4 for aPDR patients, as well as 44.5 ± 9.6 for healthy controls.

### 3.2. Cytokine Levels in the Plasma in DR and Healthy Controls

sCD121a and MCP-1 were detectable in plasma, while IL-6 and IL-8 were below the detection limit (10 pg/mL). Evaluation of sCD121a or soluble Interleukin 1 receptor type I (sIL-1R1) using CBA revealed a significant increment of sCD121a in plasma of T1D DR patients compared to healthy controls and in mNPDR compared to healthy controls (*p* = 0.004 and 0.01, respectively, Figure 2). Plasma levels of MCP-1 did not statistically significantly differ between groups (neither between healthy controls and T1D DR nor between heathy controls and mNPDR, mNPDR and aPDR or aPDR and iPDR Appendix A Table A1).

### 3.3. Cytokine Levels in PBMCs Supernatants

Levels of IL-1α, IL-1β, IL-6, IL-8, IL-10, IL-17A, IFN-γ, MCP-1 and TNF-α in supernatants of PBMC cultured with and without LPS stimulation were measured using CBA. IL-1α, IL-1β, IL-6, IL-8, IL-10, MCP-1 and TNF-α were detectable while IL-17A and IFN-γ were below the detection limit (10 pg/mL).

#### 3.3.1. Cytokine Levels in PBMC Supernatants between DR and Healthy Control

PBMC from DR patients secreted significantly higher levels of IL-10 compared to healthy controls in both the non-stimulated and LPS-stimulated conditions (Table 2). There was no difference in the levels of other cytokines investigated between DR and healthy controls (Table 2).

#### 3.3.2. Early DR: Cytokine Levels in PBMC Supernatants between mNPDR and Healthy Control

Compared to healthy controls, PBMC from mNPDR secreted significantly increased levels of IL-10 in both the non-stimulated and LPS-stimulated conditions while the secretion of IL-1α was significantly increased in the non-stimulated condition (Table 2). The levels of other cytokines investigated did not differ between the two groups (Table 2).

#### 3.3.3. Advanced DR: Cytokine Levels in PBMC Supernatants between mNPDR and aPDR

Compared to mNPDR, in the non-stimulated condition, PBMCs from aPDR secreted a significantly lower level of both IL-6 and IL-10. The difference disappeared after LPS stimulation (Table 2). There was no difference in other cytokines investigated between the two groups (Table 2).

#### 3.3.4. Cytokine Levels in PBMC Supernatants between iPDR and aPDR

Under normal culture conditions, PBMCs from aPDR and iPDR produced comparable levels of cytokines (i.e., IL-1α, IL-1β, IL-6, IL-8, IL-10, MCP-1 and TNF-α, Appendix A Table A2). LPS treatment significantly increased all cytokines measured in our study although the level of IL-6 was significantly higher in the supernatants from aPDR compared to that from iPDR (Appendix A Table A2).

### 3.4. Altered Intracellular Cytokine Expression in PBMCs in DR

#### 3.4.1. PBMC Intracellular Cytokine Expression in DR vs. Healthy Controls

Differences between DR and healthy controls were detected in two cytokines, IL-17A and IFN-γ. IL-17A was detected in small populations of CD3 T cells and CD11b myeloid cells whereas IFN-γ was only detected in CD3 T cells (Table 3). In the non-stimulated condition, DR patients compared to healthy controls showed a significant increase in the percentages of IL-17A^+^ myeloid cells and IFN-γ^+^ PBMCs (Table 3). However, the geometric mean fluorescent intensity (MFI) of IFN-γ on CD3^+^ T cells was significantly reduced in DR (Table 3). The expression of IL-17A in myeloid cells from DR patients was also significantly higher than that from healthy controls after HG, D-mannitol or LPS treatment (Table 3). The MFI of IFN-γ^+^CD3 T cells from DR patients was significantly lower than that from healthy controls under different treatment conditions (HG, 25 mM D-mannitol, LPS, or PMA + ionomycin, Table 3). We did not detect any statistically significant difference between DR and healthy controls in the expression of IL-4, IL-6, IL-8 and IL-10 in PBMCs in the non-stimulated condition (Appendix A Table A3).

#### 3.4.2. Early DR: PBMC Intracellular Cytokine Expression in mNPDR vs. Healthy Controls

IFN-γ and IL-6 were found to be differently expressed in PBMCs between the mNPDR and healthy controls (Table 4). IL-6 was detected in 0.5~1% of CD3 T cells and 21~30% of CD11b cells (Table 4). In the non-stimulated condition, compared to the healthy controls, the mNPDR patients showed a significant decrease in the percentages of IL-6^+^CD3^+^ T cells and IFN-γ^+^PBMCs and the MFI of IFN-γ^+^CD3^+^ T cells (Table 4). After LPS stimulation, the percentage of IL-6^+^CD3^+^ T cells in the mNPDR patients remained lower than that in the healthy controls, so was the percentage of IFN-γ^+^CD3^+^ cells under D-mannitol and PMA + ionomycin treatment conditions (Table 4). The MFI of IFN-γ^+^CD3^+^ T cells from mNPDR patients was significantly lower than that from healthy controls under all treatment conditions including high glucose, mannitol, PMA and ionomycin, and LPS treatment conditions (Table 4).

We did not detect any statistically significant difference between mNPDR and healthy controls in the expression of other cytokines including IL-4, IL-8, IL-10 and IL-17A in PBMCs in the non-stimulated condition (Appendix A Table A4), although the percentage of IL-4^+^CD3^+^ T cells was reduced in mNPDR patients compared to healthy controls after HG and LPS stimulation (Appendix A Table A4).

#### 3.4.3. Advanced DR: PBMC Intracellular Cytokine Expression in aPDR vs. mNPDR

IL-6 was differently expressed in PBMCs between aPDR and mNPDR. In the non-stimulated condition, the overall percentages of IL-6-expressing PBMCs, IL-6^+^CD3^+^ T cells and IL-6^+^CD11b^+^ cells alongside the MFI of IL-6 on CD11b^+^ cells were significantly higher in aPDR compared to mNPDR (Table 5). HG treatment further increased the percentage of IL-6-expressing PBMCs, particularly IL-6^+^CD11b^+^ cells in aPDR compared to that in mNPDR. The treatment also increased IL-10 production in aPDR compared to mNPDR, particularly in CD3^+^ T cells (Appendix A Table A5). The percentages of CD3^+^IL-6^+^ T cells were significantly increased in aPDR compared to mNPDR upon D-mannitol, LPS or PMA + ionomycin treatment (Table 5).

We did not detect any significant difference between aPDR and mNPDR in the expression of IFN-γ, IL-4, IL-8, IL-10 and IL-17A in PBMCs in the non-stimulated condition (Appendix A Table A5). Additionally, in the non-stimulated condition, aPDR did not show any statistically significant differences with iPDR although PBMCs from iPDR had reduced IL-10 and IFN-γ expression compared to aPDR (Appendix A Table A6). All PBMC cytokine expression results for iPDR patients are provided in Appendix A Table A6.

### 3.5. T1D Duration Is Associated with Increasing CD11b^+^IL-6^+^ PBMCs

A statistically significant positive association was found between diabetes’ duration and the percentages of CD11b^+^IL-6^+^ PBMCs in unstimulated culture condition (*p* = 0.04, R^2^ = 0.122, B = 0.827; see Figure 3). These results suggest that a longer duration of disease leads to a higher level of PBMC-derived IL-6, contributing to DR pathology. No other significant associations were found between diabetes’ duration and evaluated cytokine parameters including IFN-γ^+^ PBMCs and IFN-γ MFI on CD3^+^ PBMCs; IL-6^+^ and CD3^+^IL-6^+^ PBMCs and IL-6 MFI on CD11b^+^ PBMCs; myeloid IL-17A^+^ PBMCs; CD11b^+^ and myeloid PBMCs.

## 4. Discussion

Our results suggest that T1D DR PBMCs, particularly CD11b myeloid cells, are proinflammatory, and the increased IL-6, IL-10 and IL-17A derived from them may contribute to the retinal pathology in DR. A decreased level of PBMC-derived IFN-γ was detected, which may contribute to the diminished adaptive immunity in T1D DR patients [19]. Our observations are consistent with those of our previous study [19] where we found the proinflammatory immunophenotype in T1D DR patients is the result of an enhanced innate response and an impaired adaptive response. Our data also suggest that the IL-1/CD121a pathway may contribute to the early stages of DR.

Regarding cytokine secretion from PBMC, a higher level of IL-10 appears to be related to T1D DR pathology and may be involved in the early stages of DR as it was increased predominantly in mNPDR but less so in aPDR. Higher levels of IL-1α and IL-6 were also observed in mNPDR compared to healthy controls, suggesting that they may contribute to early T1D DR. The likely sources of the increase in secreted IL-6 and IL-10 related to early T1D DR are myeloid PBMCs. In the advanced stage of DR, the secretion of IL-6 and IL-10 by PBMCs was decreased, suggesting that this reduced secretion may be involved with the active proliferation of vessels in DR.

Concerning intracellular cytokine expression profiles in T1D DR, the increase in IL-17A^+^ PBMCs, especially CD11b^+^IL-17A^+^ cells and decrease in CD3^+^IFN-γ^+^ cells appear to be critical factors. Decreased IFN-γ and IL-6 expression by CD3 T cells is likely to be involved in the early stages of T1D DR. Meanwhile, higher intracellular IL-6 expression by CD11b^+^/myeloid cells and CD3^+^ T cells is related to the advanced stages of T1D DR. These data have value in helping us to understand how the immune system may contribute to DR, especially at the early and advanced stages of DR. Activated immune cells may participate in retinal vascular/neuronal degeneration by (1) releasing inflammatory cytokines into the plasma. Consequently, these cytokines circulate throughout the body and increase to a threshold where they damage endothelial cells; (2) releasing inflammatory cytokines which directly affect other cells that are in close contact with them. For example, leukostasis of CD11b^+^ cells [25] may release IL-17A to adjacent endothelial cells leading to BRB damage. In this case, locally produced IL-17A is sufficient to damage endothelial cells even though systemic levels of IL-17A may remain unchanged; (3) releasing “protective” or anti-inflammatory cytokines as part of a compensatory immune mechanism that may lead to repair. Consequently, this anti-inflammatory mechanism becomes dysfunctional, leading to an imbalance between pro-inflammatory and anti-inflammatory immune mechanisms that may allow DR to progress to an advanced stage.

IL-17A is a proinflammatory cytokine involved in the augmentation of immune response by prompting increased production of other proinflammatory cytokines including IL-6 and IL-1β, thereby creating a connection between T-cells’ activation and inflammation, implicating it in various autoimmune diseases [26,27]. IL-17A can be produced by Th17 T cells [26], γδ T cells and NK cells [28,29]. Our data suggest that the increase in IL-17A^+^ PBMCs may be involved in T1D DR, which is in line with findings by Honkanen et al. [30], wherein the authors reported increased IL-17 expression and secretion in T1D PBMCs. Systemic levels of IL-17A are significantly elevated in the serum of patients with T2D DR [31] and in T1D DR [32] as well as in PBMCs of T2D patients with no DR [8] compared to controls. We recently reported that IL-17A can directly damage the BRB via the JAK1 signaling pathway [33]. Data from the current study and the literature suggest that IL-17A may be a critical player in sustaining DR pathology.

In DR, the plasma, serum [12,14], vitreous [15,16] and PBMCs of DR patients express higher concentrations of IL-6 [34]. We found IL-6-producing CD11b^+^ cells were significantly increased in aPDR and the level was positively correlated with T1D duration. Diabetes’ duration is a clinical risk factor for DR and our data suggest that T1D duration has a positive correlation with PBMC-derived IL-6 which may promote DR pathology. The increased IL-6 intracellular expression was paired, however, with a decreased IL-6 secretion in aPDR. This suggests PBMC IL-6 accumulation and defective IL-6 trafficking/signaling in the aPDR stage. It has been suggested that the IL-6 secretion bottleneck is mostly due to defective IL-6 transcription and translation [35]; however, the IL-6 trafficking process is not fully understood, and it is unclear whether the increase in IL-6 synthesis with decrease in IL-6 trafficking into the microenvironment is due to deficiencies in the post-Golgi trafficking pathways that mediate IL-6 release or insufficient external cellular stimulation to prompt cytokine release. Remarkably, IL-6 is an important activator of signal transducer and activator of transcription 3 (STAT3) [36,37]. We have previously shown [38] that in T1D DR, there are higher levels of activated STAT3 in circulating leukocytes of patients with mNPDR but not in aPDR, which may be explained by current findings.

IL-10 plays an immunosuppressive role by inhibiting IL12 and TNF-α production [39,40] and is mainly produced by Th2 cells [41]. Although IL-10 is considered to be anti-inflammatory, especially on monocytes due to its suppression of production and secretion of proinflammatory cytokines such as IL-12, IL-8 and TNF-α [42], IL-10 can induce various signaling that initiate or sustain inflammation [36,37]. In agreement with our studies, reports have shown that IL-10 production is elevated in T1D patients compared to controls [32,43] as well as in T1D diabetic nephropathy. Increased secretion of IL-10 by PBMCs in the early stages of DR may be a compensatory immune mechanism that confers protection on the retina. However, our data suggest that after an extended period of increased anti-inflammatory activity of IL-10, as DR advances, there is a reduction in PBMC-secreted IL-10 which may contribute to neovascular processes in aPDR. The exact cause for reduced PBMC secretion of IL-10 in advanced DR is unclear, but it is likely that the PBMCs that release IL-10 may become impaired, allowing the advancement of DR.

IFN-γ is mainly produced by Th1 cells after their differentiation in response to pathogenic infections, driving cell-mediated immunity and stimulating B-cells for opsonizing antibody production [44]. Irregular IFN-γ expression is linked with autoinflammatory and autoimmune diseases [45]. The diminished IFN-γ observed in DR patients especially in mNPDR patients aligns with Foss et al. [43] and may shed more light on diminished adaptive immunity, susceptibility to infections and non-resolution of low-grade inflammation in diabetic patients [19,43,46,47].

We found increased levels of secreted IL-1α in supernatant of mNPDR PBMCs compared to controls. We also found that sCD121a or sIL-1R1, an inflammatory receptor to IL-1α, IL-1β and IL-1Ra [48] is increased in the plasma of T1D DR, and in mNPDR compared to controls. The binding actions of CD121a account for IL-1 mediated inflammation and pathology [48,49]; loss of IL-1 signaling in CD121a^−^/^−^ mice attenuates the formation of acellular capillaries in diabetic retina [50] and infiltration of pro-inflammatory leukocytes into ischemic tissue [51]. Our results suggest that increased IL-1α secretion and plasma levels of sCD121a may be involved in T1D DR by activating IL-1-mediated inflammation. sCD121a may be released as a protective response during the early stages of DR to counteract the noxious effects of IL-1α, and IL-1/CD121a signaling may partially explain early retinal changes including capillary acellularity in DR.

The strengths of this study include its prospective study design, the detailed phenotyping of patients and, importantly, the masking of laboratory researchers to the precedence of the blood samples (i.e., whether they had been obtained from people with DR or from healthy controls). The additional strengths of the study include (1) the use of different stimuli to induce cytokine production in PBMC and (2) the detection of intracellular cytokines. This study design allowed us to understand how the immune cells would respond under disease conditions and the key cellular sources of disease-causing cytokines (e.g., increased IL-6 and IL-17 production by CD11b cells and impaired IFN-γ production by CD3 cells in DR). Limitations include the cross-sectional study design, small patient numbers in each of the DR subgroups and the recruitment of all patients from a single site (Belfast, Northern Ireland). Additionally, it is difficult to determine what size of change in the different data presented may be mechanistically important and clinically relevant, independently of whether statistically significant. Furthermore, we detected differences in values among the different individuals studied. This interindividual variability, however, is not surprising since it is expected that pathogenic pathways may not affect all individuals in the same manner. Even within “homogeneous” groups of people with DR based on ETDRS severity, for example, many patients will reach the moderate non-proliferative DR stage but only few will develop DMO or PDR. Furthermore, some patients may develop DMO and never progress to PDR and vice versa; some will suffer both, others neither. Thus, is it likely that the differential weight of pathogenic pathways among people with T1D DR may explain subtle subject-to-subject variability in phenotype and differences in onset, progression and even complications of DR.

## 5. Conclusions

This study demonstrates that T1D DR PBMCs show proinflammatory phenotypes that are consistent with enhanced innate responses, specifically enhanced PBMC-released IL-10, IL-6 and in particular IL-17A production from myeloid cells, and impaired adaptive responses, specifically impaired CD3 T cell IFN-γ production. A longer duration of diabetes was associated with higher levels of IL-6 by CD11b^+^ cells which suggests a relationship with the disease and their potential to be used as biomarkers of DR. Further studies may shed light on the mechanisms sustaining these proinflammatory phenotypes in T1D DR and discern potential new pathways for therapeutic intervention.

## Figures and Tables

**Figure 2 cells-12-00467-f002:**
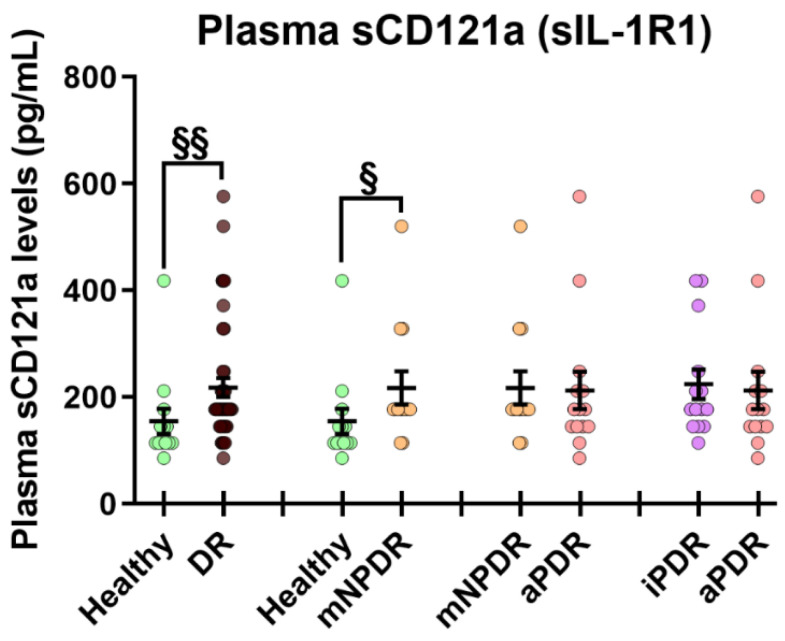
sCD121a (sIL-1R1) expression in plasma of DR patients and controls. Results are presented as mean ± SEM. *n* = 13–14 per group except DR where *n* = 41. ^§^
*p* < 0.05, ^§§^
*p* < 0.01 in Mann–Whitney U test.

**Figure 3 cells-12-00467-f003:**
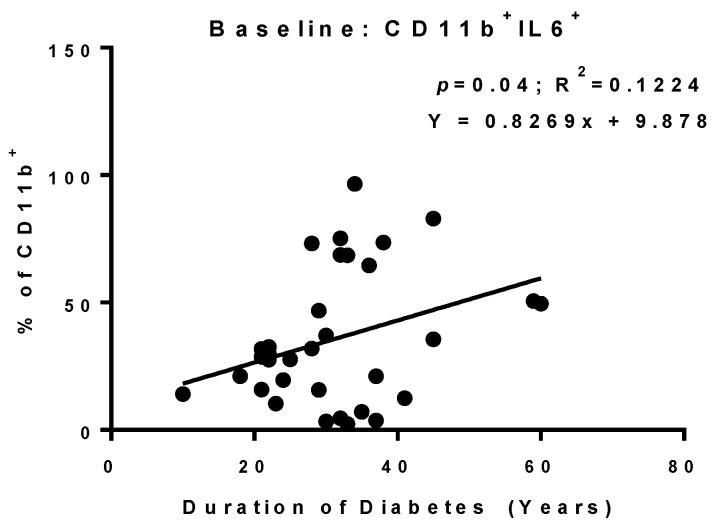
The relationship between duration of type-1 diabetes (T1D) and cytokine expression of PBMCs of DR patients. Correlation of T1D duration against percentages of CD11b^+^IL-6^+^ PBMCs in non-stimulated condition. Regression line, significance levels (*p*), coefficient of determination (R^2^) and regression equation (*y* = *a* + *bx*) are shown on each graph. *n =* 41. Linear regression.

**Table 1 cells-12-00467-t001:** Antibodies for flow cytometry and conditions for immunostaining of peripheral blood mononuclear cells (PBMC).

Antigen	Clone	Conjugate	Dilution	Reactivity	Source	Catalogue
CD3	UCHT1	FITC	1:40	Human	BD Biosciences	555332
CD11b	ICRF44	APC-Cy7	1:40			557754
IL-6	MQ2-13A5	APC	1:40		eBioscience	17-7069-42
IL-17A	eBio64CAP17	PE	1:20			12-7178-42
IFN-γ	4S.B3	APC-Cy7	1:40		Biolegend	502530
IL-10	JES3-9D7	BV421	1:40			501421
Rat IgG, κ	RTK2071	BV421	1:40	Isotype control	Biolegend	400429
Mouse IgG, κ	MOPC-21	APC-Cy7	1:40			400127
Mouse IgG, κ	MOPC-21	PE	1:20			400139
Rat IgG, κ	eBRG1	APC	1:40		eBioscience	17-4301-81

Abbreviations: APC, allophycocyanin; APC-Cy7, allophycocyanin and cyanine 7; BV, brilliant violet; FITC, fluorescein isothiocyanate; PE, phycoerythrin.

**Table 2 cells-12-00467-t002:** Cytokine levels of supernatants from PBMCs with or without LPS stimulation.

Variables (pg/g)	Healthy Controls (*n* = 13)		DR (*n* = 41)		mNPDR (*n* = 13)		aPDR (*n* = 14)	
	(Mean ± SD)							
	Base	LPS	Base	LPS	Base	LPS	Base	LPS
**IL-1α**	0 ± 0	27.67 ± 28.21	1.24 ± 4.59	37.00 ± 42.83	2.77 ± 7.14 ^*b^	38.78 ± 46.35	0 ± 0	39.77 ± 36.87
**IL-1β**	28.95 ± 49.55	1048 ± 561.6	73.07 ± 169.3	1052 ± 722.7	132.8 ± 265.6	1168 ± 710.4	25.37 ± 67.02	1113 ± 698.5
**IL-6**	846.8 ± 852.6	8738 ± 2398	1373 ± 1677	10417 ± 6262	1969 ± 1987	10100 ± 5556	**692.2 ± 849.8 *^c^**	11956 ± 5101
**IL-8**	9693 ± 5958	33946 ± 8256	10301 ± 7148	39609 ± 19218	11890 ± 7849	39201 ± 20382	9044 ± 6769	44103 ± 16459
**IL-10**	0 ± 0	41.09 ± 25.80	**6.956 ± 15.97 *^a^**	**121.2 ± 143.0 *^a^**	**15.1 ± 23.25 *^b,^ *^c^**	**149.7 ± 195.9 *^b^**	0.52 ± 1.802	113.5 ± 115.2
**MCP-1**	1868 ± 1674	2879 ± 1625	3106 ± 3716	4288 ± 3631	4149 ± 4468	4026 ± 3992	1929 ± 2076	5326 ± 4223
**TNF-α**	56.15 ± 61.2	668.0 ± 398.2	132.0 ± 205.1	890.3 ± 909.2	183.2 ± 246.5	853.70 ± 580.7	71.41 ± 154.2	838.4 ± 831.2

**Bold** shows statistically significant differences on comparison with another participant group under the same culture conditions. *****
*p* < 0.05 in Independent Samples *t*-test. ^a^ HC vs. DR, ^b^ HC vs. mNPDR and ^c^ mNPDR vs. aPDR. aPDR, active PDR; Base (Baseline); DR, diabetic retinopathy; LPS, *E. coli* lipopolysaccharide; mNPDR, mild non-proliferative DR; SD, standard deviation.

**Table 3 cells-12-00467-t003:** Cytokine expression of PBMCs from DR patients and healthy controls in different culture conditions.

Cytokine	Variable	Base		HG		Man		LPS		P + I	
		Control	DR	Control	DR	Control	DR	Control	DR	Control	DR
		Mean ± SD									
**IL-17A**	IL-17A^+^ (% of PBMCs)	1 ± 0.59	1.48 ± 1.24	1.14 ± 0.91	1.98 ± 2	0.91 ± 0.6	1.5 ± 1.4	0.713 ± 0.3	**1.45 ± 1.28 ^*^**	1 ± 0.5	1.77 ± 1.6
	CD3^+^IL-17A^+^ (% of CD3^+^)	0.79 ± 0.46	0.97 ± 1.16	0.85 ± 0.62	1.1 ± 1.77	0.76 ± 0.41	0.96 ± 1.14	0.71 ± 0.29	0.94 ± 1.18	0.67 ± 0.29	0.91 ± 1.06
	IL-17A on CD3^+^ (MFI)	135.7 ± 22.94	143.1 ± 44.45	135.1 ± 23.54	144.3 ± 45.62	137.2 ± 23.71	144.4 ± 46.29	137.2 ± 22.72	146.4 ± 47.37	133.7 ± 24.72	144.3 ± 48.46
	Myeloid IL-17A^+^ (% of myeloid PBMCs)	2.11 ± 1.5	**3.96 ± 4.2 ^†^**	2.14 ± 0.96	**5.52 ± 5.98 ***	2.19 ± 1.37	**5.2 ± 4.95 ***	2.28 ± 1.35	**5.37 ± 4.91 ****	2.44 ± 1.69	5.35 ± 6
	IL-17A on CD11b^+^ (MFI)	285.3 ± 55.77	310.7 ± 47.53	278.8 ± 63.76	**318.3 ± 55 ***	295.6 ± 63.7	317.8 ± 50.95	137.2 ± 22.72	146.4 ± 47.37	282.3 ± 61.8	318.5 ± 59.17
**IFN-γ**	IFN-γ^+^ (% of PBMCs)	1.39 ± 0.44	**1.79 ± 3.2 ^†^**	1.45 ± 0.56	1.25 ± 1.09	1.71 ± 1.47	1.17 ± 0.8	1.29 ± 0.46	1.28 ± 1.69	1.7 ± 1.04	**1.11 ± 0.69 ***
	CD3^+^IFN-γ^+^ (% of CD3^+^)	1.43 ± 0.86	1.13 ± 1.03	1.37 ± 0.75	1.05 ± 0.83	1.59 ± 0.89	**0.99 ± 0.68 ***	1.33 ± 0.75	1.24 ± 1.47	1.53 ± 0.61	**1.03 ± 0.81 ^††^**
	IFN-γ on CD3^+^ (MFI)	37.19 ± 6.12	**29.54 ± 33.21 ***	36.33 ± 6.17	**28.76 ± 8.63 ****	38.35 ± 7.69	**29.19 ± 9.13 ****	35.74 ± 6.64	**29 ± 9.12 ***	38.34 ± 8.2	**28.82 ± 9.21 ****

**Bold** shows statistically significant differences on comparison of DR patients and healthy controls under the same treatment conditions. *****
*p* < 0.05, ******
*p* < 0.01 in Independent Samples *t*-test, while **^†^**
*p* < 0.05, **^††^**
*p* < 0.01 in Mann–Whitney test. Base (Baseline); DR, diabetic retinopathy; HG, D-glucose (25 mM); LPS (*E. coli* lipopolysaccharide); Man, D-mannitol (25 mM); MFI, mean fluorescent intensity; P + I; phorbol 12-myristate 13-acetate and ionomycin; SD, standard deviation; *n* = 14 for controls and 39 for DR.

**Table 4 cells-12-00467-t004:** Cytokine expression of PBMCs from mNPDR patients vs. healthy controls in different culture conditions.

Cytokine	Variable	Base		HG		Man		LPS		P + I	
		Control	mNPDR	Control	mNPDR	Control	mNPDR	Control	mNPDR	Control	mNPDR
		Mean ± SD									
**IFN-γ**	IFN-γ^+^ (% of PBMCs)	1.39 ± 0.44	**1.09 ± 0.88 ^†^**	1.45 ± 0.56	1.03 ± 0.54	1.71 ± 1.47	1.16 ± 0.77	1.29 ± 0.46	1.12 ± 0.82	1.7 ± 1.04	**1.01 ± 0.65 ^†^**
	CD3^+^IFN-γ^+^ (% of CD3^+^)	1.43 ± 0.86	0.95 ± 0.57	1.37 ± 0.75	0.92 ± 0.36	1.59 ± 0.89	**0.9 ± 0.46 ***	1.33 ± 0.75	0.97 ± 0.61	1.53 ± 0.61	**0.87 ± 0.4 ****
	IFN-γ on CD3^+^ (MFI)	37.19 ± 6.12	**29.57 ± 8.42 ***	36.33 ± 6.17	**28.59 ± 6.72 ****	38.35 ± 7.69	**29.32 ± 8.7 ***	35.74 ± 6.64	**29.06 ± 8.15 ^*^**	38.34 ± 8.2	**27.25 ± 7.25 ****
**IL-6**	IL-6^+^ (% of PBMCs)	2.506 ± 1.99	2.26 ± 3.21	3.93 ± 3.99	2.84 ± 3.55	3.34 ± 2.81	2.75 ± 3.33	2.5 ± 2	1.93 ± 1.9	2.26 ± 1.64	2.6 ± 3.27
	CD3^+^IL-6^+^ (% of CD3^+^)	0.80 ± 0.1866	**0.61 ± 0.27 ***	1.06 ± 0.53	0.73 ± 0.36	0.77 ± 0.51	0.58 ± 0.12	0.9 ± 0.21	**0.54 ± 0.29 ****	0.89 ± 0.31	0.68 ± 0.18
	IL-6 on CD3^+^ (MFI)	28.82 ± 10.42	32.88 ± 28.42	29.58 ± 10.29	27.87 ± 12.66	28.72 ± 10.31	29.76 ± 12.04	29.82 ± 9.37	27.53 ± 10.44	28.97 ± 9.99	28.9 ± 14.18
	CD11b^+^IL-6^+^ (% of CD11b^+^)	23.88 ± 16.28	21.34 ± 25.96	29.73 ± 21.31	26.15 ± 23.18	30.14 ± 19.89	29.02 ± 22.39	27.66 ± 16.49	26.88 ± 20.98	21.35 ± 15.09	25.31 ± 24.38
	IL-6 on CD11b^+^ (MFI)	52.64 ± 68.38	98.35 ± 336.4	82.87 ± 106.8	90.13 ± 202.3	72.65 ± 93	102.7 ± 245.1	61.3 ± 64	73.19 ± 121.8	35.44 ± 68.51	88.92 ± 210.7

**Bold** shows statistically significant differences on comparison of mNPDR patients and healthy controls of the same culture conditions. *****
*p* < 0.05, ******
*p* < 0.01 in Independent Samples *t*-test while **^†^**
*p* < 0.05 in Mann–Whitney test. Base (Baseline); HG, D-glucose (25 mM); LPS (*E. coli* lipopolysaccharide); Man, D-mannitol (25 mM); MFI, mean fluorescent intensity; mNPDR mild non-proliferative diabetic retinopathy; P + I; phorbol 12-myristate 13-acetate and ionomycin; SD, standard deviation; *n* = 14 for controls and 13 for mNPDR.

**Table 5 cells-12-00467-t005:** Cytokine expression of PBMCs from aPDR vs. mNPDR patients in different culture conditions.

Cytokine	Variable	Base		HG		Man		LPS		P + I	
		mNPDR	aPDR	mNPDR	aPDR	mNPDR	aPDR	mNPDR	aPDR	mNPDR	aPDR
		Mean ± SD									
**IL-6**	IL-6^+^ (% of PBMCs)	2.26 ± 3.21	**6.78 ± 6.11 ***	2.84 ± 3.55	**6.38 ± 5.65 ***	2.75 ± 3.33	7.16 ± 6.9	1.93 ± 1.9	4.44 ± 4.04	2.6 ± 3.27	6.18 ± 6.05
	CD3^+^IL-6^+^ (% of CD3^+^)	0.61 ± 0.27 ^†^	**1.25 ± 1.24 ^†^**	0.73 ± 0.36	1.14 ± 0.6	0.58 ± 0.12	**1.39 ± 1.2 ****	0.54 ± 0.29	**1.11 ± 0.69 ***	0.68 ± 0.18	**1.23 ± 0.71 ***
	IL-6 on CD3^+^ (MFI)	32.88 ± 28.42	30.17 ± 6.36	27.87 ± 12.66	30.21 ± 5.1	29.76 ± 12.04	30.66 ± 5.9	27.53 ± 10.44	29.92 ± 5.34	28.9 ± 14.18	30.41 ± 4.39
	CD11b^+^IL-6^+^ (% of CD11b^+^)	21.34 ± 25.96	**47.93 ± 25.8 ***	26.15 ± 23.18	**46.78 ± 25.22 ***	29.02 ± 22.39	50.58 ± 29.26	26.88 ± 20.98	44.13 ± 26.8	25.31 ± 24.38	43.39 ± 28.85
	IL-6 on CD11b^+^ (MFI)	98.35 ± 336.4	**171.7 ± 159.4 ^†^**	90.13 ± 202.3	154.6 ± 155.9	102.7 ± 245.1	204.6 ± 226.9	73.19 ± 121.8	179.7 ± 291.1	88.92 ± 210.7	138 ± 152

**Bold** shows statistically significant differences on comparison of mNPDR patients and healthy controls of the same treatment conditions. *****
*p* < 0.05, ******
*p* < 0.01 in Independent Samples *t*-test while **^†^**
*p* < 0.05 in Mann-Whitney test. aPDR, active proliferative diabetic retinopathy; Base (Baseline); HG, D-glucose (25 mM); LPS (*E. coli* lipopolysaccharide); Man, D-mannitol (25 mM); MFI, mean fluorescent intensity; mNPDR mild non-proliferative diabetic retinopathy; P + I; phorbol 12-myristate 13-acetate and ionomycin; SD, standard deviation; *n* = 13 for mNPDR and 14 for aPDR.

## Data Availability

All data presented in this study are available on request from the corresponding author.

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
