# Peer review of "Peripheral Blood Mononuclear Cells from Patients with Type 1 Diabetes and Diabetic Retinopathy Produce Higher Levels of IL-17A, IL-10 and IL-6 and Lower Levels of IFN-γ—A Pilot Study"

_cells, 2023, doi:10.3390/cells12030467_

Round 1

Reviewer 1 Report

Summary: The authors Obasanmi et al. conducted a study in 41 age and gender matched type 1 diabetic patients investigating the alterations in inflammatory cytokines in the peripheral mononuclear cells. Intracellular cytokine expression, Cytokine production in plasma, non-stimulated and LPS-stimulated PBMC supernatant were assessed. The authors concluded that, they observed an increased intracellular expression of IL-17A and reduced expression of IFN- γ expression in CD3+ cells. The authors states that, uncovering this information is critical, not only to improve understanding the pathophysiology of diabetic retinopathy, but also to design more effective therapies capable of normalizing the altered immune response underpinning diabetic retinopathy 

Comments: 

  1. The study is very detailed.  
  2. However, the number of subjects recruited in the study is small. 
  3. Could the authors include a paragraph on the novelty of the current study compared to the previously conducted studies? 
  4. How many male and female subjects were included in each group of this study (i.e., mild non-proliferative DR , active pro-liferative DR (aPDR) , and PDR that have received laser PRP . 

Author Response

Summary: The authors Obasanmi et al. conducted a study in 41 age and gender matched type 1 diabetic patients investigating the alterations in inflammatory cytokines in the peripheral mononuclear cells. Intracellular cytokine expression, Cytokine production in plasma, non-stimulated and LPS-stimulated PBMC supernatant were assessed. The authors concluded that, they observed an increased intracellular expression of IL-17A and reduced expression of IFN- γ expression in CD3+ cells. The authors states that, uncovering this information is critical, not only to improve understanding the pathophysiology of diabetic retinopathy, but also to design more effective therapies capable of normalizing the altered immune response underpinning diabetic retinopathy 

Comment 1: The study is very detailed. However, the number of subjects recruited in the study is small. 

Response: We acknowledged in the Discussion of the manuscript ( (Section 4, Lines 429-431) that the small sample size of this pilot study was a study limitation. However, the study has a number of strengths (detailed in our response to comment #2, below),   We believe that, nonetheless, our study provides foundational information for further investigations of the mechanistic contributions of circulating immune cells and specific cytokines released by them to the different stages of T1D DR.

Comment 2: Could the authors include a paragraph on the novelty of the current study compared to the previously conducted studies? 

Response: Previous studies examined the number/population of PBMC and reported altered cell number/population or cell surface molecule expression in innate and adaptive immune cells in people with DR. Previous studies also reported cytokine production in serum or plasma of people with diabetes and DR. The current study further this knowledge by investigating the function (e.g., their cytokine production in response to different stimuli) of PBMC from DR patients and examined the cellular sources of different inflammatory cytokines in different conditions.

In order to address these issues, we have added the following information to the Discussion section of the reviewed manuscript, highlighting the strengths and novelties of this study:

Additional strengths of the study include (1) the use of different stimuli to induce cytokine production in PBMC, and (2) the detection of intracellular cytokines. This study design allowed us to understand how the immune cells would respond under disease conditions and the key cellular sources of disease-causing cytokines (e.g., increased IL-6 and IL-17 production by CD11b cells and impaired IFNγ production by CD3 cells in DR).

Comment 3: How many male and female subjects were included in each group of this study (i.e., mild non-proliferative DR, active proliferative DR (aPDR), and PDR that have received laser PRP . 

Response: We thank the reviewer for this important comment. The current study is an extension of our earlier study, where the detailed demographics (including gender distribution) and clinical characteristics of study participants were presented  (https://doi.org/10.1080/02713683.2020.1718165).

Section 3.1, Lines 192-193 of the manuscript currently reads as: “Detailed information on characteristics of the participants has been published elsewhere [19].

For full table, see Table 2 of Obasanmi et al. Current eye research. 2020 Sep 1;45(9):1144-54. DOI: https://doi.org/10.1080/02713683.2020.1718165 ; Authors’ copy (no paywall): https://pureadmin.qub.ac.uk/ws/portalfiles/portal/199053376/DR_Inflammation_EER_Accepted_version.pdf

However, in other to address this issue, we have added to the manuscript in Section 3.1, Lines 193-200: Briefly, females represented 29.6% of the total 54 recruited subjects, including 12 female diabetic retinopathy (DR) patients (22.2% of DR patients) with a breakdown of four patients with mNPDR (30.8% of mNPDR patients), five patients with intermediate diabetic retinopathy (35.7% of iPDR patients), and three patients with aPDR (21.4% of aPDR patients), as well as four healthy female controls (30.8% of healthy controls). The mean age ± standard deviation (years) for all subjects was 47 ± 11.7, including 47.8 ± 12.27 for DR patients, 47.8 ± 9.2 for mNPDR patients, 41.9 ± 12.5 for iPDR patients, and 53.6 ± 12.4 for aPDR patients, as well as 44.5 ± 9.6 for healthy controls.”

Reviewer 2 Report

This is a study aiming at assessing the composition of the population of PBMCs from healthy donors compared to T1D patients with different stages of DR. The differentiation between mild NPDR and PDR patients, either active or post-treatment is rather interesting but the findings are relatively minimal with very modest and largely variable effects on secretion levels of cytokines and some subpopulations of isolated PBMCs.

Major comments

Presentation of Figure 2 could be improved to help with clarity and visualization of the individual values. Rather than using different shapes, using smaller puncta of different colors would be clearer. This is important because of the lack of significance of any difference except for the “whole group” of DR, and to a very limited extent the mNPDR. Also, please clarify why the DR group for this analysis only has 39 values instead of 41 (14+14+13)? 

Please refrain from using statements such as “appeared to be increased” when not statistically different and when the differences are rather limited such as those for MCP1. There is obviously way too much variability to reach any conclusion. Please rephrase.

Statistical analysis needs to be double checked with a biostatistician as there is some inconsistencies as to the test used for comparison of multiple groups. In Table 3 for example, some of the differences are reported based on T-test which is wildly inappropriate. 

Overall, the changes observed are rather modest and with very high variability, which requires added information as to the median and the individual values to be able to assess if the changes are reflecting only a couple of samples driving said changes or more of a population change. In other words, individual values are required to be presented for at least the “main” observation to allow the reader to visualize if this is a general impact or a more subtle change affecting a sub-group of the populations tested. 

Other comments           

There is inconsistency in the way supplementary data are referred to (Appendix vs Supplementary table, i.e. A1 vs S1). Please fix. 

Author Response

Major comments

Comment 1: Presentation of Figure 2 could be improved to help with clarity and visualization of the individual values. Rather than using different shapes, using smaller puncta of different colors would be clearer. This is important because of the lack of significance of any difference except for the “whole group” of DR, and to a very limited extent the mNPDR. Also, please clarify why the DR group for this analysis only has 39 values instead of 41 (14+14+13)? 

Response: We thank the reviewer for this suggestion to improve the presentation of our data. We have followed the advice of the reviewer and have improved the Figure (page 6)

Regarding the number of DR subjects for this analysis, we have corrected this error in Line 213. We successfully measured sCD121a (sIL-1R1) in all 41 patients included in this study.

Comment 2: Please refrain from using statements such as “appeared to be increased” when not statistically different and when the differences are rather limited such as those for MCP1. There is obviously way too much variability to reach any conclusion. Please rephrase.

Response: We thank the reviewer for this suggestion. We have edited the manuscript in Section 3.2, Lines 207-210 to read as:

“Plasma levels of MCP-1 did not statistically significantly differ between groups (neither between healthy controls and T1D DR nor between heathy controls and mNPDR, mNPDR and aPDR or aPDR and iPDR (Appendix Table A1).”

Comment 3: Statistical analysis needs to be double checked with a biostatistician as there is some inconsistencies as to the test used for comparison of multiple groups. In Table 3 for example, some of the differences are reported based on T-test which is wildly inappropriate.

Response:

We thank the reviewer for this important comment. Our aim was to understand how PBMCs from healthy control and DR respond differently to inflammatory or oxidative insults. We have discussed the data with our biostatistician. Our original tables 3-5 and table A3-6 were misleading. To better demonstrate our study aim, we have now revised the tables with the control and DR next to each other for each of the treatment conditions. This would justify the use of t-tests for each of the pairwise comparison. More specifically, T-test was used to compare parametric continuous data and Mann Whitney test was used to compare the data that didn’t meet parametric conditions.

We have revised Tables 3-5 and Tables A3-A6 to effectively communicate our study's aim and results. These tables are now presented in a comprehensive and compelling format that illustrates the full extent of our findings.

Comment 4: Overall, the changes observed are rather modest and with very high variability, which requires added information as to the median and the individual values to be able to assess if the changes are reflecting only a couple of samples driving said changes or more of a population change. In other words, individual values are required to be presented for at least the “main” observation to allow the reader to visualize if this is a general impact or a more subtle change affecting a sub-group of the populations tested. 

Response: We thank the reviewer for raising these points. We provided mean and standard deviation (SD) in all the tables, and the SD is a measure of the dispersion of data. Indeed, it is difficult to determine what is the size of the change that may be mechanistically important and clinically relevant, independently on whether or not statistically significant. With regard to the  interindividual variability, in our opinion it is not surprising that this exist and does not necessarily mean that the results are less valid as a result, since it is expected that pathogenic pathways may not affect all individuals in the same manner, even within “homogeneous” groups of people with DR based on ETDRS severity,  For example, it is clear that many patients will reach the moderate non-proliferative DR stage but only few will develop DMO or PDR.  Furthermore, some patients may develop DMO and never progress to PDR and vice versa; some will suffer both.  Thus, is it likely that the differential weight of pathogenic pathways among people with T1D DR  may explain subtle subject-to-subject variability in phenotype and differences in onset, progression and even complications of DR.

 To address the reviewer’s concern, we have added additional limitations to the manuscript in Section 4, Lines 431-442 of the manuscript, as follows:

“Also, it is difficult to determine what size of change in the different data presented may be mechanistically important and clinically relevant, independently on whether or not statistically significant. Furthermore, we detected differences in values among different individuals studied. This, interindividual variability, however, is not surprising since it is expected that pathogenic pathways may not affect all individuals in the same manner, even within “homogeneous” groups of people with DR based on ETDRS severity, for example, it is clear that many patients will reach the moderate non-proliferative DR stage but only few will develop DMO or PDR.  Furthermore, some patients may develop DMO and never progress to PDR and vice versa; some will suffer both, others neither.  Thus, is it likely that the differential weight of pathogenic pathways among people with T1D DR may explain subtle subject-to-subject variability in phenotype and differences in onset, progression and even complications of DR.”

 Other comments           

There is inconsistency in the way supplementary data are referred to (Appendix vs Supplementary table, i.e. A1 vs S1). Please fix. 

Response: We thank the reviewer for this observation. In the current manuscript, we consistently refer to the supplementary data in the format ‘Appendix Table Ax’ in Lines 201-202, 235, 238, 253, 275, 277, 291, 295-296, 298-299.

Round 2

Reviewer 2 Report

The authors have addressed the comments appropriately.